# Making sense of and working with COVID-19 related guidelines and information in Danish general practice–A qualitative study

**Julie Høgsgaard Andersen**, **Anne Møller, Tina Drud Due** *

The Research Unit for General Practice and Section of General Practice, Department of Public Health, University of Copenhagen, Copenhagen, Denmark

* tina.due@sund.ku.dk

## Abstract

### Background

Attempts to manage the COVID-19 pandemic have involved a massive flow of guidelines and information to health professionals on how to reorganize clinical work and handle patients with COVID-19. The aim of this paper is to investigate how Danish general practitioners (GPs) made sense of and worked with guidelines and associated information on COVID-19 in the first months of the pandemic.

### Methods

We conducted qualitative interviews with 13 GPs in the beginning of the pandemic and again approximately three months later. Between the two interviews, they wrote daily notes for 20 days. Interviews were audio-recorded and transcribed, and the material was analyzed using thematic network analysis.

### Results

The interviewed GPs found the situation urgent and serious, and they spent a lot of time reading and working with COVID-19 related guidelines and associated information. Keeping up-to-date with and implementing guidelines was challenging due to the many sources of information and the constant guideline revisions. The GPs were able to assess patients' risk status but were challenged by the changing guidelines regarding this. The GPs found that deciding whether a COVID-19 patient needed to be admitted to hospital was relatively straightforward. An important final challenge was discrepancies between the government's public announcements regarding which patients could be tested for COVID-19, the guidelines provided to GPs, and the local testing capacities, which gave GPs extra work.

### Conclusion

In an urgent situation like the COVID-19 pandemic it is crucial to secure good communication between the government, health authorities, professional medical societies, and health professionals. Improved practices of collaboration between health authorities and

**Data Availability Statement:** According to Danish GDPR policies we cannot make the interview data available. The interviewees where promised anonymity and confidentiality and we do not

believe it is possible to provide transcripts of the interviews in a way that provide sufficient information to analyze the data and also maintain anonymity. As stated in the PLOS ONE policy we have excerpts of the transcripts in the paper in the form of quotes. Parts of anonymised transcribed interviews in Danish are available from the first author upon reasonable request. A non-author institutional contact for data access: Frans Boch Waldorff, Professor, The Research Unit for General Practice and Section of General Practice, Department of Public Health, University of Copenhagen E-mail: fransw@sund.ku.dk.

**Funding:** The participating GPs were remunerated for the interviews with a grant from The Committee of Multipractice Studies in General Practice (Grant number: EMN: 2020.01302-1388655).

**Competing interests:** The authors have declared that no competing interests exist.

**Abbreviations:** GP, General practitioner; DSAM, Dansk Selskab for Almen Medicin [The Danish College of General Practitioners; PLO, Praktiserende Lægers Organisation [The Organization of General Practitioners; NBH, The National Board of Health (Sundhedsstyrelsen; SSI, Statens Serum Institut. SSI is under the auspices of the Danish Ministry of Health. Their main duty is to ensure preparedness against infectious diseases and biological threats as well as control of congenital disorders in Denmark; PPE, Personal protective equipment.

professional societies could improve communication in future health crises and relieve GPs of some of the work involved in keeping up-to-date with information flows, constantly reviewing new guidelines, and dealing with communicative inconsistencies.

## Introduction

In Denmark, the first patient with COVID-19 was detected late February 2020. The number of infected citizens rose rapidly thereafter, and on 11 March the government announced the first lockdown of the Danish society [1]. In the following weeks and months attempts to manage and mitigate the pandemic in Denmark involved a massive flow of guidelines and associated information from the government and health authorities aimed both at health care professionals and the general public.

General practice is a cornerstone in the Danish health care system and the first point of contact for patients with health concerns. All Danish citizens are registered with a general practice, providing free and universal access to primary healthcare. The political Organisation of general practitioners (Praktiserende Lægers Organisation (PLO)) and the regional health authorities make agreements about remuneration of the GPs every third year. Each general practitioner (GP) has between 1600 and 2000 patients on his list, and GPs can be organised in larger partnerships or work solo. During the pandemic, GPs had to take care of their patients' regular health problems and help limit the spread of COVID-19. To do so, general practice took upon it a massive reorganization of clinical work and practical procedures in the very beginning of the pandemic. This reorganization was based on guidelines coming from health authorities and on associated information from the general practitioners' professional societies [2–4]. To our knowledge, there are no other studies of how general practitioners worked with COVID-19-related guidelines and information in Denmark or elsewhere.

We know from the research literature that implementation of clinical guidelines faces challenges on several levels. In their review, Fischer et al. identify personal, guideline-related, and external factors that can act as barriers to implementation. Personal factors concern physicians' knowledge about and attitudes towards a guideline. Guideline-related factors concern the evidence and plausibility of the guideline, as well as its accessibility and applicability. Finally, external factors concern organizational constraints, such as time restrictions and heavy workload [5]. We know from implementation research more broadly that sustainable implementation necessitates that professionals understand what they are supposed to do, are able to do it, and find it meaningful in relation to their professional role and the institutional context [6].

During the first wave of the COVID-19 pandemic in Denmark, guidelines and associated information were published unusually fast and based on an uncertain and constantly changing knowledge base, to meet the urgency of the situation. The aim of this paper is to examine how Danish GPs made sense of and worked with guidelines and associated information on COVID-19 in the first months of the pandemic. With the pandemic still among us, it is crucial to obtain knowledge on how key actors in the health care system experience communication from the authorities regarding the pandemic. This knowledge will also be useful in the handling of future pandemics, if an immediate reorganization of clinical work is required.

## Setting

The Danish National Board of Health (NBH) develops the official guidelines for the management of relevant known diseases in the health care sector, and of pandemic outbreaks such as

COVID-19. The first main guideline was published 15 January 2020, and by 27 May 2021 it had been updated 24 times [2]. Most updates were made in March and April 2020. The main guideline provides the background for the guideline, knowledge on epidemiology and conditions for infection, the course of the disease and people in increased risk, diagnostic assessment, indications for diagnostic assessment and test for COVID-19, management of people (suspected of) having COVID-19, and employees in the health care sector, elder care, and certain parts of the social sector [2]. In addition to the main guideline, several supplementary guidelines have been published and continuously updated by NBH [7], covering for example the planning of activities and prevention of spread of COVID-19 in the health care system; personal protective equipment (PPE); identification and management of close contacts, and guidelines for the vaccination programme. The GPs' professional societies The Danish College of General Practitioners (DSAM, Dansk Selskab for Almen Medicin), which is the scientific college of GPs, and PLO have continuously alerted GPs when NBH issued new guidelines. Furthermore, DSAM and PLO have published specific information and guidance on how to implement the guidelines. For example, DSAM created a list suggesting which patients to see in the clinic and which patients to offer a telephone or video consultation [8]. In collaboration with the five Danish administrative regions, PLO have sent out information e.g. on the financial agreement on remuneration for the use of video and extended telephone consultations, which was made in March 2020, and tips on how to get started with video consultations [4]. Before the pandemic, there were no fee for video consultations, as such consultations were not used in general practice, and only a fee for short telephone consultations. In addition, the regional departments of PLO have sent out information to general practitioners about the management of COVID-19 within their region, e.g. status on local testing capacity, supply of PPE, and where to send children who need to be tested. In this paper, we use 'guidelines' to refer to guidelines coming from NBH and 'information' to refer to information from DSAM and PLO.

The Danish Government and the NBH have mainly informed the public through the media, signs and billboards in public space, and NBH's webpage. The Government has held several press conferences with representatives from NBH, Statens Serum Institut (SSI) and the police force, where they have given a status on the spread of the pandemic and presented the restrictions put in place in order to control it.

## Methods

### Design and data collection

This was a qualitative study involving 13 GPs (see Table 1). JHA has a Ph.D. and is an experienced qualitative health researcher with a background in anthropology. AM has a Ph.D. and is both actively working as a GP and a senior researcher. Her Ph.D. from 2013 was a mixed method study and included qualitative work. TDD is an experienced qualitative health service and implementation researcher with a background in public health. All authors participated in the design of the study. To accelerate the recruitment process and obtain data in the initial phase of the pandemic convenience sampling was used: AM together with other GP colleagues at the Research Unit for General Practice in Copenhagen contacted GPs through posts in GP Facebook groups, and by asking GPs in their networks. We strove to reach variation among the participants in terms of practice type, gender, and seniority. The participants had clinics in either Region Zealand or The Capital Region of Denmark. Eight participants were women. Four GPs had a solo practice, while the rest were in shared practices of differing sizes (see Table 1). This reflect the wide variation in how Danish general practices are organized, with

**Table 1. GP characteristics.**

|  | Practice type | GPs in the practice | Region | Gender | Age | Years as GP |
|---|---|---|---|---|---|---|
| 1 | Partnership | 3 GPs | Region Zealand | Female | 55–59 | 15–19 |
| 2 | Partnership | 4 GPs | Region Zealand | Male | 45–49 | 10–14 |
| 3 | Partnership | 4 GPs | Region Zealand | Female | 45–49 | 5–9 |
| 4 | Partnership | 2 GPs | Region Zealand | Female | 45–49 | 0–4 |
| 5 | Solo | 1 GP | Region Zealand | Female | 50–54 | 10–14 |
| 6 | Partnership | 5 GPs | Capital Region of Denmark | Female | 60–64 | 25–29 |
| 7 | Partnership | 2 GPs | Capital Region of Denmark | Female | 55–59 | 20–24 |
| 8 | Partnership | 3 GPs | Capital Region of Denmark | Female | 60–64 | 20–24 |
| 9 | Solo | 1 GP | Capital Region of Denmark | Male | Unknown | Unknown |
| 10 | Solo | 1 GP | Region Zealand | Male | 70–74 | 25–29 |
| 11 | Solo | 1 GP | Region Zealand | Female | 60–64 | 25–29 |
| 12 | Partnership | 5 GPs | Region Zealand | Male | 50–54 | 10–14 |
| 13 | Partnership | 9 GPs | Region Zealand | Male | 55–59 | 20–24 |

some practices consisting of just one GP and a secretary while others contain several GPs and staff such as nurses, nurse assistants, and young doctors in training.

The GPs were interviewed twice, and in each round JHA and TDD respectively conducted half of the interviews using video-calls to avoid risk of COVID-19. We conducted the first round of interviews in the beginning of April 2020. After having carried out 13 interviews, we assessed that no new issues were raised in the last few ones. Hence, we concluded that these were sufficient for the study. We conducted a second round of interviews with eleven of these GPs at the end of June. A twelfth of the GPs was interviewed at the end of August, and the last GP declined a second interview for personal reasons. Interviews lasted between 1 and 1.5 hours. The themes of the interview guides are presented in Table 1. The interview guides were semi-structured and though fairly consistent they were adjusted a few times during the first interviews to elaborate on perspectives uncovered in the initial interviews. To get an impression of how daily work in the clinic was affected by the pandemic, the participants were asked to make written or audio-recorded daily notes for 20 days after the first interview, taking a point of departure in the four questions in Table 2. Eleven of the GPs provided notes. The interviewees were informed about the purpose of the study at recruitment and again at the beginning of each interview. In the following, the notion "the GPs" refers to the GPs in this study whereas "GPs" refers to the general population of Danish GPs.

## Analysis

JHA and TDD conducted an explorative and inductive thematic network analysis of the interviews [9]. We started with a reading of the entire material to familiarize ourselves with the data and to develop an initial coding framework. Hereafter, we both coded the same two interviews to compare our use of the codes, which resulted in a few changes in the coding framework. We then divided the remaining interviews between us and coded them using the software programme Nvivo 11 Plus. We then identified, discussed, and refined themes by going through the codes and coded extracts (from both within and across interviews). We constructed a thematic network and summarized the overall themes in a coherent narrative, which we discussed with AM. The notes from the GPs were not coded in Nvivo. Instead, we identified the main themes to see if they differed significantly from the interviews, which they did not. The notes expanded our understanding of the daily challenges

**Table 2. Interview guide and daily notes.**

| Interview themes in the first interview |
| --- |
| • Physical and organizational precautionary measures taken in the clinics to minimize the risk of spreading COVID-19 |
| • Use and experience with telephone and video consultations |
| • The GPs' perceptions of their own risk of infection and their work conditions during the pandemic |
| • Tasks and challenges related to patients with COVID-19 symptoms |
| • Information from the authorities and keeping up-to-date |
| • Work and challenges related to implementing guidelines |
| • Collaboration with the rest of the health care system |
| **Interview themes in the second interview** |
| • Follow up on issues covered in the first interview and in the notes |
| • Continued reorganisation and resumption of usual practice |
| • Use and experience with telephone and video consultations |
| • Experiences of clinical quality during the pandemic and adequacy of treatment of patients with COVID-19 symptoms and patients with regular health issues |
| • The influence of minimal use of personal protective equipment on patient care |
| • Counselling patients in increased risk of serious illness if infected with COVID-19 |
| **Questions for the written or audio recorded daily notes** |
| • Which clinical challenges have you experienced today in relation to COVID-19 and other acute/sub-acute health issues? |
| • Are you aware of any changes in clinical guidelines since yesterday? |
| • Is there anything else that has taken up time today? |
| • Do you have any suggestions as to how the situation in general practice could be improved? |

the GPs experienced during the pandemic. Through the analysis, the GPs' perception of and work with the information and guidelines coming from authorities and professional societies proved to be a dominant issue. The themes and subthemes related to this issue are presented in Table 3, and they form the structure of the results section. Another important theme was how the GPs decided between and experienced using different consultation forms. This theme is the focus of a previous article [10].

## Ethics

All participants have received written information about the project, signed an informed consent form, and been anonymized. Data are securely stored and can only be accessed by the authors. The project has been registered with the Danish Data Protection Agency (journal nr. 514-0491/20-3000), and it has been presented to the Danish ethics committees who declared that being a qualitative study, it does not need their approval (journal nr. 20023269).

**Table 3. Themes from the analysis.**

| Theme | Subthemes |
| --- | --- |
| Navigating and working with guidelines and associated information | Engagement in minimizing the spread of COVID-19 |
| | Keeping up-to-date with new guidelines and information |
| | Individual and collective work with guidelines and information |
| Using COVID-19 guidelines when advising patients | Giving advice about risk groups |
| | Managing patients with symptoms of COVID-19 |

## Results

### Navigating and working with guidelines and information

**Engagement in minimizing the spread of COVID-19.** None of our interviewees seemed to question the need to participate in minimizing the spread of COVID-19. They stated that the situation was urgent and serious, and they had to act to protect their patients and themselves and to avoid a collapse of the health care system. As one GP said

*"Now with this COVID (..) it was very serious, if we did not act properly concerning ourselves, our staff and our patients (. . .) it could have fatal consequences" (GP1-interview 1).*

Some of the GPs said that they had felt a sense of team spirit among health professionals, and that collaboration in the health care system on COVID-19 had been unusually well functioning. The GPs' statements concerning the seriousness of the situation were also reflected in the number of precautions taken to prevent spreading of COVID-19, especially following the national lockdown. These included keeping up to date with the guidelines and following their recommendations, e.g. regarding increased hygiene, removing chairs from the waiting area, transferring as many consultations as possible to video and telephone, and not seeing patients with symptoms that could be caused by corona virus. Apart from the seriousness of the situation, some of the GPs stressed that an important pre-requisite for their engagement with video- and telephone consultations and, thereby, for their possibilities of participating in disease prevention, was the agreement made regarding remuneration of video and extended telephone consultations, which helped to secure their income.

**Keeping up-to-date with new guidelines and information.** The GPs said that keeping up to date with the pandemic-related guidelines and information was challenging. First, the guidelines and the associated information were sent out by several organizations and on many platforms. They mentioned the NBH, DSAM, PLO, and the regional departments of PLO as their primary sources of knowledge. Guidelines from NBH came by e-mail, whereas DSAM and PLO sent out newsletters and posted information on their web pages and in their Facebook groups. As one GP said: *"Hails are fired from five shotguns over your head at once" (GP 7-interview 1)*. In addition to the channels of information on guidelines aimed directly at general practice, the GPs also mentioned following news media, including government press conferences. This was out of general interest, but some of the GPs also mentioned that it was necessary as changes in pandemic-related strategies, initiatives, and restrictions were sometimes introduced at these conferences before they were communicated to health professionals. This is a point we will return to below.

A second challenge to staying up-to-date, especially in March and April 2020, was that guidelines concerning COVID-19 changed daily, and that information on these updates came at diverse times during the day. Furthermore, each time a change was made to a NBH guideline, the full guideline was distributed again, at the time of the first interview this was done without highlighting the changes. Thus, the GPs spent a lot of time trying to figure out what was new each time a guideline arrived:

*"What they did in the end was that they highlighted the changes, but they did not even do that to begin with. So, you had to read the whole guideline to find out, what was new, what is changed, what is it I have to relate to now, what does my staff need to know, what do I need to know?" (GP4-interview 2)*

As explained by this GP, at the time of the second interview, such changes to existing guidelines were highlighted, and that made the GPs' work a lot easier.

At the same time, however, quite a few of the GPs said that politicians and authorities generally did well, and that some confusion is inevitable in such an unknown and chaotic situation as in the beginning of the pandemic with insufficient knowledge.

*"There is always new information and thus constantly new corrections, so I also feel that you have to be a little large and say that it is the art of the possible." (GP6-interview 1)*

**Individual and collective work with guidelines and information.** The frequency with which guidelines changed meant that the GPs' workday became longer because they used evenings and early mornings to read new, immediately applicable, guidelines and the associated information. When describing how they navigated in the flow of updates, all the GPs mentioned paying specific attention to their professional societies. The GPs explained that the guidelines coming from health authorities lacked descriptions of *how* they should be implemented, but their professional societies had done an excellent job of translating them into concrete suggestions for general practice. However, the suggestions for implementation from the professional societies necessarily came after the updated guidelines had been sent out, so while waiting for these, the GPs also had to keep an eye on the updates coming directly from the health authorities and find out how to act accordingly. As one GP explained:

*"First of all, we try to see when the emails from PLO or DSAM come, then we know, then it's something final. The problem is when it is not there, when we have a morning meeting at 10 minutes to eight. Then the evening before, you have to read the Danish Agency for Patient Safety's website, Statens Serum Institut's website, the National Board of Health's website, and try to see, well, what are the announcements written here and what can we use them for. So, clearly our focus has been on, when announcement from the societies comes, then it is definitive, because there is someone who has tried to filter the information and made it much more targeted at us." (GP2-interview 1)*

The GP further explained that in order to know exactly how guidelines could be implemented locally, they had to pay attention to the region-specific information from the regional PLO departments e.g. on testing capacity and availability of PPE.

Apart from their individual work with reading new guidelines and information, the GPs described that deciding and planning how to implement the guidelines required collective work in the clinics, and thus internal communication had intensified profoundly in the first couple of months of the pandemic. Extra meetings were held both between the GP-owners of the clinic and with the entire staff. For example, in most of the clinics, the GPs had introduced morning meetings where they discussed the latest news and decided how to respond to these:

*"We have a day structure where we all meet, all those who have to be at work, we meet 10 minutes to eight for a mini-morning meeting, where we update each other on the current status, who is sick and how do we approach the day. And there we have tried to start the meeting with what the latest recommendations are? What, related to who should be sent where?" (GP3-interview 1).*

Key topics at these meetings were for instance making sure everyone had the same perception of who could be tested for COVID-19 (as we will return to later, this was not always clear), and figuring out which patients to see in the clinic and whom to talk to on the phone or

in a video consultation. The GPs explained that they highly appreciated the list made by DSAM on who to see in the clinic. However, the list had to be interpreted according to specific patient needs. As one GP explained, it was an issue of constant internal discussion how to make this interpretation.

> *"We talk about it every single day, every single hour. And in the morning, we have a doctor's telephone hour from eight to nine, and there we typically sit five or six doctors on the phone, and the four of us can sit in the same room. So there we have the opportunity to talk together about what to do in a specific situation, and continuously reassess, and we typically also talk about it during the lunch break or during the day. And our staff is constantly asking for guidelines for what to do. And we try to give it to them in relation to what is being distributed from our societies, and then say that they should allocate the patients accordingly and confer with us, if they are in doubt." (GP6-interview 1).*

In general, the GPs were able to make sense of the guidelines. However, when reflecting on the flow of guidelines and information in the first few months of the pandemic, the GPs said that the many sources of information, the constant revisions of guidelines and the difficulties with locating what exactly had been changed in new guidelines made it almost impossible to keep up-to-date, and trying to do so left them tired and often frustrated. In the second interview, some of the GPs were fed up with reading guidelines. One said:

> *"We do not bother any more. Well, I think that most GPs are totally exhausted by now. We read and read and read in the beginning, and all of a sudden, you reach the point: "No, now we can no longer bear it. Now, we have to take it when we get the question. Then we look it up." (GP5-interview 2)*

The GPs also said that the flow of changing guidelines had diminished, as had the work needed to keep up- to -date. At this point (at the end of June 2020), the pandemic was generally well-controlled in Denmark; the incidence and prevalence of COVID-19 were low, as was the number of patients hospitalized with COVID-19 [11]. Correspondingly, large parts of the Danish society had been reopened.

## Using COVID-19 guidelines when advising patients

**Giving advice to patients about risk groups.**   Following the first phase of the reopening of society in April 2020, the GPs received many calls from patients who wanted to know if they belonged to one of the so-called risk groups, e.g. people who are in increased risk of getting severely ill if infected with COVID-19, and how they should act if they did. Patients were especially concerned about going to work. The GPs said that giving advice regarding risk groups was challenged by the often-changing guidelines.

> *"Again, there have been changing instructions, right? And when the guidelines have been changed, it has been difficult, because then we have had a few weeks before where it said: 'Now you are in the risk group. You must not, and then, next week, a new guide said: 'that actually you could if you did not have contact with directly ill people, and that sort of thing". So that part has been difficult, because it changed". (GP 3-interview 2).*

Most of the GPs said that, knowing their patients' medical situations, they could assess if they were in a risk group. However, at times it could be complicated to give advice on appropriate implications of being in a risk group for a specific patient's everyday behavior. The GPs

stressed that deciding whether a patient could go to work was not their responsibility, as that depended on the possibilities of keeping a distance to other people and the risk of infection at the workplace in general. If needed, the GPs could write a medical statement describing a patient's risk status, but it was up to the patient and his or her employer to figure out if and how the patient could show up at work. This was in line with a statement from DSAM [12]. However, a few of the GPs had talked to patients or employers who expected more direct guidance from GPs regarding which work tasks patients in risk could safely manage.

The GPs also received calls from patients who had spouses or family members who belonged to a risk group and were unsure if they should go to work or to social events. The GPs differed with respect to how manageable they found such questions. Some found them too specific to answer, while others said that they usually found some solution with the patient. Some of the GPs also found that answering such questions should not be the task of the GP; the patients should call the national corona hotline established by the authorities.

**Managing patients with symptoms of COVID-19.**   The task of managing patients with symptoms of COVID-19 changed in accordance with the development in knowledge concerning these symptoms and the testing capacity. In the beginning of the pandemic in Denmark, criteria for suspecting COVID-19 were primarily focused on having visited other countries with widespread infection or having been close to someone who had been diagnosed with COVID-19. As the pandemic spread, however, everyone was advised to stay at home if they had any illness symptoms. Until the end of March, patients could only be tested if they were so ill that they needed to be admitted to the hospital. At that time, GPs had to assess the severity of the patient's illness, which most of the interviewed GPs found manageable. However, some of them mentioned that in the very beginning of the pandemic, when knowledge was scarce about exactly which symptoms indicated COVID-19, they often conferred with the infection department at the local hospital.

From the beginning of April 2020, testing capacity gradually expanded. Across all the interviews and the daily notes, the GPs highlighted that communication regarding this expansion had been problematic. One problem was that the authorities informed the public about expansions in testing capacity during press conferences, before any specific guidelines had been issued to general practitioners about who exactly could be tested and how. Furthermore, some press conferences were held during the day, when GPs did not have the opportunity to watch them. As a result, patients started calling their GPs requesting tests before their GPs had been informed about the expansion or knew exactly what it entailed. In relation to this, the interviewed GPs found that the general statements made at the press conferences sometimes did not correspond to the actual guidelines, leading some patients to think that they could get a test when, in fact, they could not:

> "So, we are a little frustrated that now, for example, the current announcement in the press is that now everyone can be tested with mild symptoms, right. It then gets blown across the entire broadcast area, and that's actually not what the guidelines are. People hear about the guidelines in the press. They do not see what the National Board of Health announce to us, and they do not see everything that is written in small print, right. So we have to sit all the time and say: 'yes, but it's not quite like what you read in the newspaper or what you heard in the news. There are these and these limitations. And it becomes a frustration among the citizens at the other end, and we become the target for it. From the patients' viewpoint it looks as if it is something we decide, and it is not something we decide, but we try to navigate according to what we are told to do" (GP6-interview 1).

A second problem in relation to testing capacity was that the statements concerning expansions of testing capacity were made before the local possibilities for testing more people were in place. This meant some referrals to tests were rejected, and that figuring out who could be tested was complicated. For example, on 20 April 2020 the authorities announced that now anyone who had symptoms could get a test. This was not the case in Region Zealand, however, where the test centers did not have the capacity to test people with mild symptoms. The case in Box 1 shows how one GP had to make several phone calls in the days from 20–24 April, to find out which patients he could refer to a test.

---

### Box 1

On 20 April the GP writes that he has spoken to the National Board of Health's corona hotline for health professionals and the test center in Region Zealand and received the message that patients with mild symptoms cannot be tested. On 23 April PLO Zealand announces that the region is ready to test patients with mild symptoms. However, when the GP calls the regional test center he receives the message that they do not know anything, and that they are not ready to test these patients. Finally, on the 24th Region Zealand is "up to speed" with the national strategy for testing, although the GP has to wait in a telephone queue to refer his patients.

---

In addition to the problems related to testing, the GPs also described how unclear announcements concerning a rollout of pneumococcal vaccine had given them extra work. In the beginning of April 2020, the government announced that specific groups would be offered pneumococcal vaccine in order to prevent a high number of lung infections in addition to COVID-19. However, it was not made clear exactly who could get a vaccination and when. As a result, many patients started calling their GPs asking if they could get the vaccination, but since the GPs had not received a guideline with a plan for who to vaccinate, they had no answers for these patients. Such a plan was in place a couple of weeks later [13].

The GPs were very frustrated with the discrepancies between announcements, guidelines, and practical reality, as they resulted in extra work for them. Accordingly, there was a very strong plea from the GPs that they were briefed first before announcements were made to the general population, and that specific guidelines for general practice came before or at least at the same time as the announcements:

*Well, it's always nice to be ahead of things that happen, instead of being behind. So if one was to wish for anything, then it would definitely be that, before any official announcements, a briefing was planned of the groups in the health system it involved. It does not have to be several hours before, but at least at the same time. The moment that for instance a news broadcast starts, then a guideline should be distributed to healthcare staff who is affected by it. 'This is the new announcement, it will mean such and such for you. We recommend that you approach it in this way'. So, I do not want say a roadmap, but at least reflections on how they could presume it would affect our workday, and how they think we could solve it in the easiest way possible. (GP2-interview 1)*

## Discussion

### Summary of results

The interviewed GPs found the situation urgent and serious and did not question the need for reorganizing work in general practice to help prevent the spreading of the disease. The GPs spent a lot of time reading and working with guidelines and associated information on how to manage COVID-19 in general practice. Keeping up–to-date with and implementing guidelines was challenging because there were several sources of information, and because the guidelines were often revised. The GPs generally experienced that they were able to assess patients' risk status, although the constantly changing guidelines on risk groups and the assessment of implications could be challenging. In the first months of the pandemic, GPs played a major role in giving advice to patients with symptoms of COVID-19 and referring patients to tests. Deciding whether a patient should stay at home or be admitted to hospital was relatively straightforward. However, the GPs told of discrepancies between the government's public announcements regarding which patients could be tested for COVID-19, the guidelines provided to GPs and the local testing capacities. These discrepancies were considered a serious problem which gave the GPs a lot of extra work.

### Factors influencing the GPs' perception of and work with the guidelines

The dissemination of COVID-19 guidelines has been unusual in several ways: The first NBH guideline came out when knowledge of the disease was still very sparse, and revisions came with unprecedented speed due to the constant development in the knowledge base. Furthermore, the guidelines covered not only the medical work related to patients with COVID-19 but also patients in increased risk and patients with usual health problems, and they were followed by information on a multitude of practical issues related to the reorganization of clinical practice. In comparison, clinical guidelines are usually relatively limited in scope and developed over a long period of time. It is well-known from implementation research that implementation of new guidelines takes time, and several barriers and facilitating factors have been identified. In the following, we will discuss our findings in relation to barriers for guideline implementation identified in the reviews by Fischer et al. [5] and Francke et al. 2008 [14]. The COVID-19 pandemic created a setting marked by an urgency and seriousness that is unlike the settings guidelines are usually implemented in. Yet, both reviews identify barriers for guideline implementation related to personal factors, guideline related factors and external factors, which we find useful for discussing our results.

*Personal factors* include physicians' knowledge about and attitudes towards a guideline, including factors such as awareness, familiarity, agreement, motivation, outcome expectancy, skills, and self-efficacy [5, 14]. The GPs in our study received updates on new guidelines and information from many sources, and they were deeply involved in reading them all. They were motivated to do so, because they found it necessary and urgent to reorganize their daily work to help prevent spreading COVID-19. They did not have any existing knowledge of COVID-19, and they trusted NBH and their professional societies. Similarly, Houghton et al. find that a key facilitator for health care staffs' adherence to infection prevention and control guidelines for infectious respiratory diseases is that they see the value of the guideline [15]. At the time of the second interview in our study, the GPs said their motivation for spending time on reading new guidelines had decreased. But this happened at the same time as the pandemic became increasingly well-controlled in Denmark, and as fewer guideline revisions were made. It is difficult to say whether GPs would have had the energy to continue working with constantly changing guidelines if that had been necessary. The GPs generally found that they were able to

decide which patients to see in the clinic and whom to talk to using telephone and video (see also [10]). They were also able to assess the severity of patients' COVID-19 illness, and patients' risk status, even if deciding on the implications of this status could be difficult. The GPs did not see it as their job to decide whether a patient could go to work, and many said that very specific questions concerning what patients should and should not do in their everyday lives fell outside the scope of the guidelines and their area of responsibility.

*Guideline related factors* concern factors such as evidence, plausibility, complexity, layout, accessibility, and applicability [5, 14]. As already mentioned, the GPs generally trusted the content of the guidelines. However, given the many revisions of the guidelines and the many sources of information about them, reaching an understanding of exactly what GPs had to do and how involved time-consuming work. This also had to do with the guidelines' layout and applicability: In the first months of the pandemic, the changes made to the NBH guidelines were not highlighted, when updated versions were sent out. Consequently, GPs had to read the whole guideline over and over again to figure out what was new. Also, the GPs said the NBH guidelines lacked information on how to implement them. In relation to this, the GPs praised their professional societies for translating the official guidelines from NBH into concrete suggestions for implementation in general practice. According to Francke et al. [14] the most frequently described factor influencing implementation concerns complexity, so guidelines should be easy to understand and use. In line with this, Houghton et al. [15] and Kain and Jardine [16] stress the importance of short and precise risk communication in health crises such as a pandemic. In our study, working with the guidelines was a complex affair, not so much due to complexity of content but due to the many revisions and sources of information and the work needed to decide on how to implement them. Apart from reading the guidelines, the work involved intensified staff communication in the clinics, e.g. in the form of everyday morning meetings.

*External factors* involve organizational constraints and resources [5, 14]. The GPs in our study had to use their spare time to read guidelines. They found this time consuming and exhausting. Apart from that, the main external factor complicating GPs' work with the guidelines concerned discrepancies between what the government announced to the public about testing strategy, what was in the guidelines, and what was practically possible. The GPs were very frustrated about these discrepancies, as they caused them a lot of extra work taking calls from patients who thought they could be tested, when in fact they could not. Also, the GPs had to do some detective work in figuring out exactly who could be tested in their community. A positive external factor highlighted by the GPs was the agreement on remuneration of video and extended telephone consultations, which was made in the beginning of the pandemic.

Normalization process theory (NPT) is a widely used framework for understanding the implementation, embedding and integration of interventions in health care settings [6, 17]. The theory focuses on the collective, coordinated, and cooperative social action involved when actors are engaged in implementation processes [18]. The theory entails several components, of which the first two, *coherence* and *cognitive participation*, are relevant to the above discussion. Coherence concerns sense making; how actors make sense of a new practice and the work it entails for them. Cognitive participation concerns how actors engage with and enroll in a new practice. Building on this, although the GPs in our study fully supported the need to change clinical practice as a result of COVID-19, the process of making sense of the guidelines and figuring out how to implement them was an ongoing, time-consuming task. The morning meetings were thus both an occasion for discussing if everybody in a clinic had the same understanding of the guidelines and their implications, and for deciding what needed to be done and by whom. In a discussion paper on the roll out of video consultations during the COVID-19 pandemic, Bidmead and Marshall argue that *"even in the face of Covid 19, one*

*should not underestimate the importance of providing opportunities for sense making wherein staff can develop shared understandings of purpose, the potential benefits and what is expected of them which are necessarily absent in such a rapid rollout as we see in the current crisis"* [18]. Our study supports this statement by making it clear that sense making, and cognitive participation related to COVID-19 guidelines is a time-consuming endeavor in general–not only in relation to video consultations. Thus, in the future handling of the current pandemic and in future pandemics it is important to consider how guidelines can be issued so that the burden of engaging with them is minimized. This entails highlighting changes to guidelines and securing that information from the government, national health institutions, and professional societies are aligned and support each other, and that guidelines come with clear suggestions for implementation.

## Strengths and limitations

This study provides insights into how Danish GPs have made sense of and worked with the flow of COVID-19 related guidelines and information in the first few months of the pandemic. It is a strength that we interviewed the GPs twice, as that allowed us to gain knowledge on how they experienced the changes in guidelines that took place in the months between the two interviews. Furthermore, the daily notes provided real time insight into the daily frustrations the GPs experienced regarding the guidelines. We used convenience sampling by recruiting GPs through AM and our colleagues' networks in order to obtain knowledge from the start of the pandemic. This meant that we only included GPs from two regions. Including a broader sample of GPs from all of the five Danish regions might have added new insights, as there have been regional differences in the organization, e.g. of testing. Furthermore, given that the pandemic is ongoing, a third-round of interviews could have been enlightening in terms of figuring out if GPs still experience challenges related to the implementation of guidelines, or if improvements in the guidelines have been made.

## Conclusion

The interviewed GPs generally supported the national efforts to control the pandemic and the associated need to reorganize work in general practice. The many sources of information and the constant revisions of the key guidelines were the main challenges to the GPs' work with guidelines, and the discrepancies between what the government and health authorities announced to the public and what was communicated to GPs were a cause of massive frustration. The translation of the main NBH guideline into practical suggestions for practice made by the professional societies was praised by the GPs.

In a situation like the initial phase of COVID-19, it is crucial to secure good communication between the government and health authorities, professional societies and health professionals in their clinics.

Establishing improved practices of collaboration between health authorities and professional societies could improve communication in future health crises and relieve general practitioners of some of the work involved in keeping up-to-date with information flows from numerous channels, constantly reviewing new guidelines, and dealing with inconsistencies. In relation to this, it is important that information that significantly influences GPs' work is communicated to them before it is announced to the general public.

## Acknowledgments

We would like to thank all the participating GPs for their time and contributions.

## Author Contributions

**Conceptualization:** Julie Høgsgaard Andersen, Anne Møller, Tina Drud Due.

**Data curation:** Julie Høgsgaard Andersen, Tina Drud Due.

**Formal analysis:** Julie Høgsgaard Andersen, Tina Drud Due.

**Funding acquisition:** Julie Høgsgaard Andersen, Tina Drud Due.

**Investigation:** Julie Høgsgaard Andersen, Tina Drud Due.

**Methodology:** Julie Høgsgaard Andersen, Anne Møller, Tina Drud Due.

**Project administration:** Julie Høgsgaard Andersen, Tina Drud Due.

**Visualization:** Julie Høgsgaard Andersen, Tina Drud Due.

**Writing – original draft:** Julie Høgsgaard Andersen.

**Writing – review & editing:** Julie Høgsgaard Andersen, Anne Møller, Tina Drud Due.

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
