## [Decision Letter · Decision Letter 0]

5 Oct 2021

PONE-D-21-21386Making sense of and working with COVID-19 related guidelines and information in Danish general practice – a qualitative studyPLOS ONE

Dear Dr. Due,

Thank you for submitting your manuscript to PLOS ONE. After careful consideration, we feel that it has merit but does not fully meet PLOS ONE’s publication criteria as it currently stands. Therefore, we invite you to submit a revised version of the manuscript that addresses the points raised during the review process.

We look forward to receiving your revised manuscript.

Kind regards,

Imran Masood, B. Pharm, MBA, CQRM, PhD

Academic Editor

PLOS ONE

2. "When reporting the results of qualitative research, we suggest consulting the COREQ guidelines: http://intqhc.oxfordjournals.org/content/19/6/349. In this case, please consider including more information on the number of interviewers, their training and characteristics.

“We would like to thank all the participating GPs for their time and contributions.”

“The author(s) received no specific funding for this work. The participating GPs were remunerated for the interviews with a grant from The Committee of Multipractice Studies in General Practice”

Additional Editor Comments (if provided):

Reviewers' comments:

Reviewer's Responses to Questions

**Comments to the Author**

1. Is the manuscript technically sound, and do the data support the conclusions?

Reviewer #1: Yes

Reviewer #2: Yes

2. Has the statistical analysis been performed appropriately and rigorously? 

Reviewer #1: N/A

Reviewer #2: No

3. Have the authors made all data underlying the findings in their manuscript fully available?

Reviewer #1: Yes

Reviewer #2: Yes

4. Is the manuscript presented in an intelligible fashion and written in standard English?

Reviewer #1: Yes

Reviewer #2: No

5. Review Comments to the Author

Reviewer #1: The submitted paper is impressively written highlighting the importance of subject matter. Methodology chapter well defined the criteria and pathway of research. Results are well explained accordance with the findings. Discussion was thoroughly written and explained with latest references.

Reviewer #2: Comment 1.

Overall, the article is good and clear. The introduction is relevant and theory based. For this reason, I will recommend it for publication, although I have some questions that should be addressed in modifications to the text, which I include here below:

1. How you check the reliability of your interview guide? ( You didn't mention it)

2. Which sampling method is employed in this study?

3. How you concluded that 13 GP's are sufficient for this study?

4. Were the subjects ever informed of the true nature of the study. If so when and if not then why not?

Comment 2.

While the study appears to be sound, the language is unclear, making it difficult to follow. I advise the authors work with a copyeditor to improve the sentence flow and readability of the paper.

6. PLOS authors have the option to publish the peer review history of their article (what does this mean?). If published, this will include your full peer review and any attached files.

Reviewer #1: **Yes: **Dr. Saira Azhar

Associate Professor Pharmacy Practice

College of Pharmacy

University of Sargodha

Sargodha, Pakistan

0092-3076724266

saira.azhar@uos.edu.pk

Reviewer #2: No

---

## [Author Response · Author response to Decision Letter 0]

4 Dec 2021

Our point by point responses are described in below. They are also attached as a more readable table in the attached files under the name response to reviewers

Thank you. 

Comment: 

Please ensure that your manuscript meets PLOS ONE's style requirements, including those for file naming. The PLOS ONE style templates can be found at https://journals.plos.org/plosone/s/file?id=wjVg/PLOSOne_formatting_sample_main_body.pdf and https://journals.plos.org/plosone/s/file?id=ba62/PLOSOne_formatting_sample_title_authors_affiliations.pdf

Response: 

The guidelines have been read and changes have been made to comply with the requirements regarding headlines.

Comment:

When reporting the results of qualitative research, we suggest consulting the COREQ guidelines: http://intqhc.oxfordjournals.org/content/19/6/349. 

In this case, please consider including more information on the number of interviewers, their training and characteristics.

Response: 

As described in line 112-113 JHA and TDD conducted the interviews. We have added information about our characteristics in line 102-105

Comment:

We note that the grant information you provided in the ‘Funding Information’ and ‘Financial Disclosure’ sections do not match. When you resubmit, please ensure that you provide the correct grant numbers for the awards you received for your study in the ‘Funding Information’ section.

Thank you for stating the following in the Acknowledgments Section of your manuscript:

“We would like to thank all the participating GPs for their time and contributions.”

“The author(s) received no specific funding for this work. The participating GPs were remunerated for the interviews with a grant from The Committee of Multipractice Studies in General Practice”

Response:

Both ‘Funding Information’ and ‘Financial Disclosure’ sections should state: “The authors received no specific funding for this work.”

In the manuscript we submitted we cannot identify any information about funding in the Acknowledgments section. We have deleted information about remuneration in the method section. 

We received a grant solely for remuneration of the participating GPs. We are unsure where this should be stated and hope you can change this for us. 

It should state: “The participating GPs were remunerated for the interviews with a grant from The Committee of Multipractice Studies in General Practice (Grant number: EMN: 2020.01302-1388655)”

This information is also provided in the cover letter

Comment:

We note that you have indicated that data from this study are available upon request. PLOS only allows data to be available upon request if there are legal or ethical restrictions on sharing data publicly. For more information on unacceptable data access restrictions, please see http://journals.plos.org/plosone/s/data-availability#loc-unacceptable-data-access-restrictions.

Response: 

We have reread the PLOS ONE Data availability policy and believe we with this qualitative study comply with the requirements. According to Danish GDPR policies we cannot make more data available. The interviewees where promised anonymity and confidentiality and we do not believe it is possible to provide transcripts of the interviews in a way that provide sufficient information to analyze the data and also maintain anonymity. As stated in the PLOS ONE policy we have excerpts of the transcripts in the paper in the form of quotes. 

Comment: 

Your ethics statement should only appear in the Methods section of your manuscript. If your ethics statement is written in any section besides the Methods, please delete it from any other section.

Response: 

The ethics statement was supposed to be a subsection under the method section, but the level of headlines was incorrect. This has been corrected.

Comment: 

Response

The reference list has been reviewed and all webpages has been accessed again 6. November

Reviewer #1: 

Comment: 

The submitted paper is impressively written highlighting the importance of subject matter. Methodology chapter well defined the criteria and pathway of research. Results are well explained accordance with the findings. Discussion was thoroughly written and explained with latest references.

Response: 

Thank you for the nice comments.

Reviewer #2: 

Comment:

How you check the reliability of your interview guide? ( You didn't mention it)

Response: 

We have elaborated on this issue in the Design and data collection section line 119-121 where we described that the interview guides were semi-structured and though fairly consistent they were adjusted a few times during the first interviews to elaborate on perspectives uncovered in the initial interviews.

Comment: 

2. Which sampling method is employed in this study?

Response: 

We have added in line 107 to the description of the sampling that is a convenience sampling.

Comment: 

How you concluded that 13 GP's are sufficient for this study?

Response: 

We have rephrased and clarified the description hereon. 

Comment: 

Were the subjects ever informed of the true nature of the study. If so when and if not then why not?

Response: 

The interviewees were informed about the purpose of the study at recruitment and again in the beginning of the each interview. This information is added to the method section line 124-125. 

Comment: 

While the study appears to be sound, the language is unclear, making it difficult to follow. 

I advise the authors work with a copyeditor to improve the sentence flow and readability of the paper.

Response: 

The manuscript has been in language revision and changes have been made correspondingly.

---

## [Decision Letter · Decision Letter 1]

15 Dec 2022

PONE-D-21-21386R1Making sense of and working with COVID-19 related guidelines and information in Danish general practice – a qualitative studyPLOS ONE

Dear Dr. Due,

Thank you for submitting your manuscript to PLOS ONE. After careful consideration, we feel that it has merit but does not fully meet PLOS ONE’s publication criteria as it currently stands. Therefore, we invite you to submit a revised version of the manuscript that addresses the points raised during the review process.

We look forward to receiving your revised manuscript.

Kind regards,

Lucinda Shen, MSc

Staff Editor

PLOS ONE

on behalf of 

Naranjargal Dashdorj

Academic Editor

PLOS ONE

Journal Requirements:

Additional Editor Comments:

Thank you for submitting this interesting manuscript. If you can do minor edits in English that would be even better.

Reviewers' comments:

Reviewer's Responses to Questions

**Comments to the Author**

1. If the authors have adequately addressed your comments raised in a previous round of review and you feel that this manuscript is now acceptable for publication, you may indicate that here to bypass the “Comments to the Author” section, enter your conflict of interest statement in the “Confidential to Editor” section, and submit your "Accept" recommendation.

Reviewer #2: (No Response)

Reviewer #3: (No Response)

2. Is the manuscript technically sound, and do the data support the conclusions?

Reviewer #2: Partly

Reviewer #3: Yes

3. Has the statistical analysis been performed appropriately and rigorously? 

Reviewer #2: Yes

Reviewer #3: Yes

4. Have the authors made all data underlying the findings in their manuscript fully available?

Reviewer #2: No

Reviewer #3: Yes

5. Is the manuscript presented in an intelligible fashion and written in standard English?

Reviewer #2: No

Reviewer #3: Yes

6. Review Comments to the Author

Reviewer #2: First of all, the overall content is not presented in an intelligible way. Secondly, there are many errors in the sentence flow and standard english. Same error as earlier. Moreover, this manuscript findings are not clear and satisfactory. So, i am not satisfied with this manuscript as this is failed in fulfilling the requirements of PLOS. Thankyou.

Reviewer #3: Thank you for inviting me to review this paper on 'Making sense of and working with COVID-19 related guidelines and information in Danish general practice – a qualitative study'.

I really enjoyed reading it as it resonated with my own experience as a GP in England during the first few months of the pandemic.

I can see there are already some comments and responses to reviewers - please accept my apologies if this review is later than you had hoped.

The methods are transparent, the findings clear and the discussion interesting which includes drawing on existing evidence from implementation science. It may be nice to have a summary box with learning points for the reader and/or perhaps recommendations for policymakers in future.

As a reader from another country it would be really helpful to get a feel for the characteristics of the practice including how many patients they had and what staff worked in each practice (approx number and roles) - so that when reference is made to the practice meeting the reader can visualise who the GPs are meeting with.

A table with the demographics of the participants could be helpful for the reader.

The reader comes to understand that there was potentially a point in time that GPs would not be paid for a video consultation, but they eventually were - for those unfamiliar with the Danish system a couple of lines about how Danish GPs are paid would be very helpful to understand how this was even in question at some point.

I was surprised to see that ethical approval was not needed as this was a 'qualitative' study. Qualitative studies in the UK would be expected to have ethics approval. Again for the non-Danish reader a sentence explaining the logic to this would be helpful.

I noted a typo in the translation of a quote:

Pg 13, line 242 What, related to who should be send (should be sent) where?”

Pg 15 line 281. The use of the word hereon is unusual. I can see it has also been used elsewhere, and I would generally suggest removing it, as a native speaker it is not commonly used.

Well done for capturing these views during a challenging time. It was fascinating to read how much in common GPs in Denmark had with those in England.

7. PLOS authors have the option to publish the peer review history of their article (what does this mean?). If published, this will include your full peer review and any attached files.

Reviewer #2: No

Reviewer #3: **Yes: **Dr Luisa M Pettigrew

---

## [Author Response · Author response to Decision Letter 1]

26 Jan 2023

A response to reviewers is attached as a document in the resubmission

Regards Tina

---

## [Editor Report · Decision Letter 2]

27 Jan 2023

Making sense of and working with COVID-19 related guidelines and information in Danish general practice – a qualitative study

PONE-D-21-21386R2

Dear Dr. Due,

We’re pleased to inform you that your manuscript has been judged scientifically suitable for publication and will be formally accepted for publication once it meets all outstanding technical requirements.

Kind regards,

Naranjargal Dashdorj

Academic Editor

PLOS ONE
---

## [Editor Report · Acceptance letter]

2 Feb 2023

PONE-D-21-21386R2 

Making sense of and working with COVID-19 related guidelines and information in Danish general practice – a qualitative study 

Dear Dr. Due:

I'm pleased to inform you that your manuscript has been deemed suitable for publication in PLOS ONE. Congratulations! Your manuscript is now with our production department. 

Kind regards, 

on behalf of

Dr. Naranjargal Dashdorj 

Academic Editor

PLOS ONE